# Implications of a Neuronal Receptor Family, Metabotropic Glutamate Receptors, in Cancer Development and Progression

**DOI:** 10.3390/cells11182857

**Published:** 2022-09-13

**Authors:** Kevinn Eddy, Mohamad Naser Eddin, Anna Fateeva, Stefano Vito Boccadamo Pompili, Raj Shah, Saurav Doshi, Suzie Chen

**Affiliations:** 1Graduate Program in Cellular and Molecular Pharmacology, School of Graduate Studies, Rutgers University, Piscataway, NJ 08854, USA; 2Susan Lehman Cullman Laboratory for Cancer Research, Rutgers University, Piscataway, NJ 08854, USA; 3Department of Physiology and Pharmacology “V. Erspamer”, Sapienza University, 00185 Rome, Italy; 4Environmental & Occupational Health Sciences Institute, Rutgers University, Piscataway, NJ 08854, USA; 5Institute of Bioinformatics and Biotechnology, Savitribai Phule Pune University, Pune 411007, India; 6Rutgers Cancer Institute of New Jersey, New Brunswick, NJ 08901, USA

**Keywords:** cancer, guanine nucleotide binding–protein coupled receptor, metabotropic glutamate receptor, glutamate, phospholipase C, adenylyl cyclase, MAPK, PI3K/AKT, riluzole, metabolism

## Abstract

Cancer is the second leading cause of death, and incidences are increasing globally. Simply defined, cancer is the uncontrolled proliferation of a cell, and depending on the tissue of origin, the cancer etiology, biology, progression, prognosis, and treatment will differ. Carcinogenesis and its progression are associated with genetic factors that can either be inherited and/or acquired and are classified as an oncogene or tumor suppressor. Many of these genetic factors converge on common signaling pathway(s), such as the MAPK and PI3K/AKT pathways. In this review, we will focus on the metabotropic glutamate receptor (mGluR) family, an upstream protein that transmits extracellular signals into the cell and has been shown to regulate many aspects of tumor development and progression. We explore the involvement of members of this receptor family in various cancers that include breast cancer, colorectal cancer, glioma, kidney cancer, melanoma, oral cancer, osteosarcoma, pancreatic cancer, prostate cancer, and T-cell cancers. Intriguingly, depending on the member, mGluRs can either be classified as oncogenes or tumor suppressors, although in general most act as an oncogene. The extensive work done to elucidate the role of mGluRs in various cancers suggests that it might be a viable strategy to therapeutically target glutamatergic signaling.

## 1. Introduction to Cancer

Advances in human health have greatly improved the life expectancy of the human population due to the development of vaccines and antibiotics. Humans can now live well past their 60s. This improved longevity has also created new problems for the aging population. Older individuals are more susceptible to cancer and neurodegenerative diseases compared to younger individuals since they have accumulated somatic mutations and/or inherited genetic aberrations caused by exposure to environmental carcinogens (UV radiation and cigarettes) [1]. Oncogenes and tumor suppressor genes have been linked to numerous cancers; however, an oncogene in one type of cancer may act as a tumor suppressor in another cancer, suggesting the importance of not generalizing a gene as an oncogene or tumor suppressor broadly, but rather in the context of a specific cancer type [2,3,4]. There are many genetic alterations associated with oncogenes and tumor suppressors, and they converge on common signaling networks such as MAPK, PI3K/AKT, Wnt, TGFβ, Notch, and Hippo pathways [5,6,7,8]. The crosstalk among these cascades allows for cancer cells to constantly adapt to their host’s defenses by manipulating intrinsic and extrinsic biological pathways to support tumor proliferation and progression. Hanahan and Weinberg first introduced the notion of the Hallmarks of Cancer in 2000 with six distinct features that classified the progression of a normal cell to a tumor cell: sustaining proliferative signaling, evading growth suppressors, resisting cell death, enabling replicative immortality, inducing angiogenesis, and activating invasion and metastatic signals (Figure 1) [9]. In 2011, they modified the Hallmarks to include two new hallmarks and two enabling characteristics that were seen as emerging requirements of a tumor cell: reprogramming of energy metabolism, evading immune destruction, genomic instability, and inflammation (Figure 1) [10]. In 2022, an additional four Hallmarks have been added as our understanding of tumor development has improved: epigenetic reprogramming, senescence, polymorphic microbiomes, and cellular plasticity (Figure 1) [11]. In this review, we will focus on the role of metabotropic glutamate receptors (mGluR: protein; *Grm*: mouse gene; and *GRM*: human gene) family, a member of the guanine nucleotide binding–protein coupled receptor (GPCR) superfamily in carcinogenesis.

## 2. Metabotropic Glutamate Receptor Biology

The GPCR superfamily consists of receptors that share a similar overall structure and makes up nearly 4% of the human protein-coding DNA sequences (approximately 800 unique genes) [12]. Their physiological roles range from functioning in the central nervous system (CNS) to vision, hearing, taste, and the immune system [13]. There are six categories of GPCRs classified based on their sequence homology, function, ligand, and structural features. These six classes of receptors are rhodopsin-like receptors (class A), secretin receptors (class B), mGluRs (class C), fungal mating pheromone receptors (class D), cyclic adenosine monophosphate (cAMP) receptors (class E) and frizzled/smoothened receptors (class F) [14]. The rhodopsin-like receptors are the largest class in the GPCR family and make up 85% of known GPCRs [15]. Although GPCRs are diverse, they share common structural units. They all have a seven-transmembrane domain, which consists of hydrophobic residues that join the N-terminal extracellular domain and the C-terminal intracellular domain [16]. The seven-transmembrane domain has three intracellular loops and three extracellular loops that connect the alpha-helical structures of this transmembrane domain [17]. The N-terminal domain has highly conserved disulfide linkages that stabilize the receptor structure [18]. The binding of various ligands, which include neurotransmitters, lipids, ions, hormones, amines, nucleotides, and odorant molecules, can activate GPCRs. Photoactivated GPCRs can transmit extracellular photoelectrical signaling into the cell by absorbing photon energy. Once activated, the extracellular domain undergoes a conformational change causing the intracellular G-protein subunit G_α_ to exchange guanine diphosphate (GDP) for guanine triphosphate (GTP), leading to its dissociation from G_βγ_ [19]. G_α_-bound GTP directs the activation of downstream signaling cascades associated with adenylyl cyclases or phospholipase C (PLC) and others [19].

mGluRs belong to the class C GPCR family and are activated by the most abundant neurotransmitter found in the central nervous system (CNS), glutamate. mGluRs can be subdivided into groups I, II, and III based on their structure, phylogenetics, signal transduction, and pharmacology [20]. Group I consists of mGluR1 and mGluR5; group II comprises of mGluR2 and mGluR3; and mGluR4, mGluR6, mGluR7 and mGluR8 belong to group III (Figure 2) [20]. mGluRs have similar structures to other members of the GPCR family in that they have an extracellular N-terminal domain that is cysteine-rich and is attached to a seven transmembrane domain that connects to the C-terminal domain found inside the cell; however, the N-terminus has a unique feature not found in other GPCRs, known as the Venus-flytrap domain [20,21]. This domain is where glutamate binds and activates the receptor, which occurs in a similar fashion as described for GPCR signaling in general (Figure 2). The canonical function of mGluRs is in the CNS for the regulation of neuronal signaling. The three subgroups of mGluRs have distinct functions and localizations in the synapses (Figure 2) [20]. Group I mGluRs are predominantly found on the post-synapses and are responsible for neuronal excitability by stimulating the G_αq_/G_α/11_ subunits that activate PLC signaling (Figure 2) [20,22]. Group II/III mGluRs are coupled to G_αi/o_ subunits that transduce signals via inhibition of the adenylyl cyclase pathway and are localized on both the pre- and post-synapses, where they function to downregulate neuronal excitability (Figure 2) [20]. Furthermore, the dissociation of the G_βγ_ heterodimeric subunit from the G_αβγ_ heterotrimeric complex allows G_βγ_ to directly activate the G-protein, which is activated inwardly to rectify potassium channels and inhibit voltage gated calcium channels, which further contributes to the inhibitory activities of G_i/o_ in neurons [23,24,25,26,27].

Group I mGluRs signal through PLC (Figure 2) [20]. Upon the binding of glutamate to group I mGluRs, G_αq_/G_α/11_ are activated through an exchange of GDP for GTP, which then leads to G_α_ dissociation from the G_βγ_ heterodimer (Figure 2) [28]. The GTP-bound G_αq_/G_α/11_ activates PLC, leading to the cleavage of phosphatidylinositol-4,5-diphosphate (PIP_2_) to diacyl-glycerol (DAG) and inositol 1,4,5-triphosphate (IP_3_) (Figure 2) [28]. DAG remains bound to the membrane, while IP_3_ diffuses into the cytoplasm where it interacts with IP3 receptors (IP3R) on the endoplasmic reticulum to release calcium (Ca^2+^) into the cytosol (Figure 2) [28,29]. The elevated Ca^2+^ in concert with DAG stimulates protein kinase C (PKC), leading to the activation of downstream signaling pathways, such as MAPK and PI3K/AKT (Figure 2) [28]. Group II/III mGluRs signal through G_i/o_ to inhibit adenylyl cyclase, thus abrogating cAMP signaling through protein kinase A (PKA) (Figure 2) [20]. Interestingly, it has been shown that both the inter- and intra-heterodimerization of group I, II, and III mGluRs occurs with group II/III heterodimerization happening more frequently, while group I/II or I/III heterodimerization occurs in rare circumstances [30]. Group II/III heterodimerization may be more common as a result of redundancies in signal transduction pathways via the G_αi/o_ subunit, while group I/II or I/III heterodimerizations are rarer due to the incompatibility of the different G_α_ subunits involved in signaling, and the negating effect of PLC and adenylyl cyclase. The heterodimerization of groups I/II or I/III may be functionally incompatible, as they counteract each other’s functions in neuronal excitation. Group I directs neuronal excitability, while groups II/III suppress excitation. Another possibility for the heterodimerization between mGluRs may be the need to increase the diversity of mGluR signaling via the recruitment of different secondary messengers, adapter proteins, and/or scaffold proteins. An intriguing topic for future studies is the evaluation of the role of mGluR homodimerization and/or heterodimerization in cancer development and progression. There are numerous antagonists and agonists for group I–III mGluRs, which have been extensively reviewed by multiple authors; therefore, they will not be reviewed here [20,31,32,33,34].

Abnormalities in GPCRs are associated with various human health disorders that include cardiovascular disease, diabetes, bone disease, and cancer [28,35,36,37]. Our group was the first to suggest the involvement of mGluRs in cancer development and progression [38,39]. In the following sections, we will explore the role of mGluRs in various cancers.

## 3. Involvement of mGluR1 Signaling in Melanomagenesis and Progression

Our lab was the first to show the significance of mGluR1 in melanoma development and progression [38]. Using genomic DNA derived from human adipose tissue, we were able to commit embryonic mouse fibroblasts to differentiate into adipocytes in vitro; subsequently, smaller human DNA subclones were isolated from these transfectants and some were shown to have similar adipocyte differentiation activities in vitro [40,41]. Attempting to translate these in vitro findings to in vivo, transgenic mice were established with one of the DNA subclones, clone B. Five founder mice were identified and one of them, TG-3, developed pigmented lesions spontaneously at the age of 8 months; histological analysis showed that the pigmented lesions arose from transformed melanocytes [42,43]. Molecular analyses revealed that a classic case of insertional mutagenesis occurred, where clone B was inserted into the intron 3 region of the *Grm1* gene with the concurrent deletion of a 70 kb host DNA fragment [38]. To test our hypothesis that the development of melanoma was driven by the ectopic expression of *Grm1* in mouse melanocytes, a second transgenic line was created, Tg(*Grm1*)EPv, wherein the expression of *Grm1* cDNA was regulated by a melanocyte specific promoter, dopachrome tautomerase (Dct) [38]. Tg(*Grm1*)EPv mice showed similar onset and progression of melanoma as the original TG-3, supporting the involvement of mGluR1 in melanoma pathogenesis [38]. Aberrant mGluR1 expression was observed in over 65% (*n* = 175) of primary and metastatic human melanoma biopsy samples, 92% (*n* = 25) of human melanoma cell lines, and 33% of human dysplastic nevi samples with little or no expression detected in human melanocytes [38,44,45].

Normal melanocytes do not express mGluR1, and it is not known how the ectopic expression is triggered in melanoma. We speculate that the expression of mGluR1 must be tightly regulated in melanocytes, considering that the consequence of its unscheduled expression leads to the development of a metastatic cancer. Neuron-Restrictive Silencing Factor (NRSF), a Kruppel-type zinc finger transcription factor, was identified as a regulator of neuronal-specific gene expression in non-neuronal cells. NRSF interacts with the Neuron-Restrictive Silencer Element (NRSE) to suppress gene expression, which includes *Grm1/GRM1* [46,47]. Ferraguti and colleagues determined that the binding of NRSF to NRSE located 5 kb upstream of the *Grm1* initiation codon was responsible for the absence of mGluR1 expression in BHK and NIH3T3 cells [48]. Similar results were observed in human epidermal melanocytes [49]. On the epigenetic level, the demethylation of the *GRM1* promoter region was found to also play a role in the aberrant expression of mGluR1 [49]. In addition to NRSF/NRSE interaction and epigenetic regulation, the Sp1 transcriptional activator was also found to be involved in the regulation of *GRM1* in human melanoma cells [49].

We and others have shown that the sustained expression and function of mGluR1 are necessary for the maintenance and progression of melanoma tumors in vitro and in vivo [39,45,50,51]. mGluR1-expressing melanomas were demonstrated to have elevated levels of glutamate in their tumor microenvironment, established by an autocrine/paracrine loop that allows for the hyperactivation of the receptor, leading to the stimulation of the oncogenic MAPK and PI3K/AKT pathways in a manner that is independent of *BRAF* and *NRAS* mutations, as well as mGluR5 expression/activity (Figure 3) [28,39,45,50,52,52,53,54]. The mGluR1-mediated activation of these pathways leads to increased melanoma cell proliferation, angiogenesis, invasion, and metastasis, as well as resisting cell death, enabling replicative immortality, avoiding immune destruction, and dysregulating cellular metabolism [28]. We have shown that the MAPK cascade is directly regulated by mGluR1, while PI3K/AKT signaling is likely regulated via the transactivation of insulin-like growth factor receptor 1 (IGF-R1) through the mGluR1 activation of Src [52,53,55,56]. The mGluR1-mediated upregulation of the PI3K/AKT/mTOR/HIF-1α pathway promotes increased expression of angiogenic factors, such as vascular endothelial growth factor (VEGF) and interleukin-8 (IL-8)—molecules shown to be responsible for the vasculature expansion required for tumor growth and metastasis to distal sites [57].

Recently, deubiquitinase cylindromatosis (CYLD), a well-known tumor suppressor, was demonstrated to be another player in melanoma progression and invasion. The downregulation of CYLD is controlled by the transcription factor Snail1, which leads to an increase in levels of Cyclin D1, and N-cadherin, promoting tumor proliferation and invasion [58]. Bosserhoff and colleagues showed that a homozygous loss of CYLD led to a shortened latency in melanoma development and progression in our *Grm1*-driven spontaneous melanoma mouse model Tg(*Grm1*)EPv, as well as increased vasculogenic mimicry and lymph angiogenesis [59]. Another group found CYLD to be a regulator of NF-κB, a transcription factor that promotes cell survival and oncogenesis [60]. We have shown that NF-κB is constitutively active in our mGluR1-expressing melanoma cells, suggesting the potential involvement of mGluR1 in the CYLD–NF-κB axis [61]. We speculate that NF-κB signaling is regulated by mGluR1 through downregulation of tumor suppressor CYLD to promote melanoma growth and metastasis; however, further investigation is needed. A recent report proposed that the loss of CYLD in melanoma cells leads to the upregulation of euchromatic histone-lysine N-methyltransferase 2 (EHMT2), which is associated with the di-methylation of H3K9 and heterochromatin formation and contributes to increased tumor proliferation [62]. The SOX10 transcription factor is crucial for proper melanocyte differentiation and survival. It is required for mGluR1 driven melanoma development and progression both in vitro and in vivo [63]. However, MITF, a melanocyte lineage determining factor, was shown not to be essential for mGluR1-driven melanomagenesis, although it is a downstream effector of SOX10, as well as of KIT, a receptor tyrosine kinase [63,64,65]. The study was performed with crosses between heterozygous knockout MITF mice and our Tg(*Grm1*)EPv mice, because homozygous MITF knockout mice lack melanocytes [63]. It will be interesting to use an inducible MITF mouse model to explore the relationship between mGluR1 and MITF once melanocytes have been established. Altogether, SOX10 may be involved in mGluR1-driven melanomagenesis independently of its effector, MITF [63].

The possibility that *GRM1*/mGluR1 may be uniquely qualified to serve as a viable target for melanoma therapies was assessed in vitro and in vivo with riluzole (Rilutek^®^), an FDA-approved treatment for Amyotrophic Lateral Sclerosis (ALS) [66,67]. One of the known functions of riluzole is inhibiting glutamate release, thereby decreasing glutamate levels in the extracellular space, which results in a reduction in mGluR1 mediated signaling (Figure 3) [39]. The treatment of mGluR1-expressing melanoma cells with riluzole significantly reduced glutamate release into conditioned cultured media, which resulted in a reduction in cell proliferation/viability and an increase in apoptosis (Figure 3) [39]. Riluzole’s impact is specific to mGluR1-expressing melanomas, as normal melanocytes and mGluR1-negative melanoma cells are unaffected [39,68]. Subsequently, we showed that riluzole treatment reduced allografted or xenografted tumor progression in vivo with no obvious liver toxicities [39]. Based on our pre-clinical data, we translated our results to a proof of principle phase 0 single agent riluzole trial in late-stage melanoma patients using the riluzole dose approved for ALS patients [66]. A significant decrease in FDG-PET signals was detected in 34% of the patients, and a reduction in MAPK and PI3K/AKT signaling cascade was observed after only 14 days of dosing in paired specimens [66]. These exciting results led to a phase 2 therapeutic clinical trial with the single agent riluzole [67]. No complete response was observed, but 46% of the patients showed stable disease with concurrent reduction in MAPK and PI3K/AKT signaling cascade activation, and an increase in infiltration of lymphocytes within the stromal/tumor junctions only in post-treatment stable disease patient samples [67]. These results suggest that some patients may benefit from single-agent strategies; however, most of the late-stage melanoma patients are unlikely to experience a long-lasting benefit [67].

The investigation of riluzole’s mechanism of action revealed that majority of riluzole-treated cells are arrested at the G_2_/M phase in the cell cycle within 24 h, with subsequent apoptosis after 48 h (Figure 3) [39]. The high proportion of cells arrested at the G_2_/M phase signifies possible DNA damage—a notion that was verified by increased levels of reactive oxygen species (ROS) and γ-H2AX, a marker of DNA double-stranded breaks, in melanoma cells treated with riluzole (Figure 3) [69]. We confirmed these observations with post-treatment melanoma biopsies obtained from our completed phase II trial [66,69]. We speculate that riluzole inhibits the export of intracellular glutamate via the glutamate/cystine antiporter, xCT, which results in a decrease in the import of cystine [69,70]. This correlated with diminishing of cysteine available to participate in glutathione synthesis, thus yielding a rise in ROS levels, as evident by the elevated levels of γ-H2AX (Figure 3) [69,70]. Further studies have shown that cells with the double-stranded DNA breaks caused by riluzole treatment had reduced efficiency in DNA damage repair by the nonhomologous end joining (NHEJ) repair pathway (Figure 3) [71]. As such, we propose that riluzole disrupts the xCT antiporter, resulting in the interruption of the cell detoxification mechanisms (Figure 3) [69]. Several investigators have reported on the anti-proliferative activities of riluzole in gliomas; however, no further studies have been performed to examine if, in the presence of riluzole, the cells exhibit DNA damage [72,73,74]. A recent report by Mahajan and colleagues demonstrated increased DNA damage in riluzole-treated osteosarcoma cells [75]. Furthermore, oxidative stress has been proposed to be a component of many neurodegenerative diseases; however, it is not known if in ALS or other neurodegenerative diseases, patients treated with riluzole show elevated levels of DNA damage [76,77,78]. Based on the results of our phase II riluzole monotherapy trial, as shown previously in ALS patients, there were high interpatient variabilities in the bioavailability of riluzole due to the inconsistent expression of the hepatic enzyme cytochrome P450 isoform CYP1A2 that metabolizes riluzole [67,79]. To overcome this, troriluzole, a prodrug of riluzole, was developed, which allows for a uniform exposure of riluzole to be achieved across all patients regardless of CYP1A2 expression. Currently, our lab is investigating the preclinical and clinical efficacy of the combined treatment of troriluzole with anti-PD1, an immune checkpoint inhibitor, in melanoma patients with brain metastasis (NCT03229278 and NCT04899921). For an in-depth review on mGluR1 signaling in melanoma development and progression, readers should refer to the review written by Eddy and Chen titled “Glutamatergic Signaling a Therapeutic Vulnerability in Melanoma” [28].

## 4. The Role of Metabotropic Glutamate Receptors in Various Cancers

In this section, we will briefly review studies that have shown the role of mGluRs in different cancers, including neuronal, breast, kidney, prostate, colorectal, gastrointestinal, melanoma, ovary, and upper aerodigestive tract cancers [22,80,81,82,83,84,85,86,87,88,89,90,91,92,93,94,95,96,97,98,99,100].

### 4.1. Breast Cancer

mGluR1 was reported to play a role in tumorigenesis and progression in human breast cancers. The introduction of *Grm1* cDNA into immortalized Mouse Mammary Epithelial Cells (iMMECs) resulted in cellular transformation in vitro and tumorigenesis in vivo, with enhanced angiogenesis [101]. mGluR1-expressing iMMECs showed elevated levels of extracellular glutamate, and inclusion of a glutamate release inhibitor, riluzole, reduced cell growth in vitro and tumor progression in MCF7 xenografts in vivo [101]. mGluR1 expression might be a good prognostic marker in predicting patient survival in estrogen receptor (ER)-positive, ER-negative, and triple-negative breast cancers (TNBC) [102,103,104]. Speyer and colleagues demonstrated that in the presence of riluzole, mice inoculated with 4T1, an mGluR1-expressing breast cancer cell line, displayed a reduction in tumor growth and blood vessel formation [105]. Furthermore, mGluR1 signaling is required for endothelial cell (EC) development and blood vessel formation [105]. These results indicate that elevated levels of glutamate in mGluR1-expressing cancer cells lead to the unintended consequence of angiogenesis induction due to the activation of mGluR1 receptors on ECs (Table 1). In addition, Speyer and co-workers showed that mGluR1 regulates acute inflammation in TNBCs by upregulating the cytokines CXCL1, IL-6, and IL-8 [106]. These data suggest that mGluR1-expressing cancer cells could use cytokine production to regulate immune surveillance and immune cell infiltration to allow tumor cells to evade immune detection via the modulation of mGluR1 intra-tumoral signaling, i.e., prevent immune cells from eliciting an anti-tumor immune response (Table 1). Taken together, combining pharmacological inhibitors of mGluR1 signaling with immunotherapies could be a rational therapeutic approach in breast cancer.

Another member of the mGluR family, *GRM8*, was shown to function as an oncogene in breast cancer and was linked to a worse overall survival rate (Table 1) [107]. The abnormal expression of mGluR8 in breast cancer cells led to increased cell proliferation, migration, invasion, tumorigenesis, and inhibition of cell death signaling [107]. It was demonstrated that *GRM8* is negatively regulated by miR-33a-5p in breast cancer [107]. Others have also linked *GRM4* overexpression in MDA-MB-231, a human breast cancer cell line, to a reduction in cell proliferation, migration, and invasion [108]. In contrast, the knockdown of *GRM4* promoted these activities (Table 1) [108].

Further investigations are needed to clearly determine whether one or more members of the mGluR family could be a reliable target in breast cancer. Taken together, the available data suggest that various members of the mGluR family have differential impacts on different breast cancer types.

### 4.2. Colorectal Cancer

In colon cancer, mGluR4 is overexpressed compared to normal colon cells [100,109]. Functional assays revealed that mGluR4 signaling enhanced tumor cell proliferation and invasion [100]. It was observed that 68% of human colorectal adenocarcinomas overexpressed mGluR4, which is correlated with a worse prognosis and poor disease-free survival, suggesting mGluR4′s role as an oncogene (Table 1) [100]. The overexpression of mGluR4 was shown to contribute to 5-fluorouracil resistance (a chemotherapy drug) as compared to non-resistant parental cells [100,109]. Previously proposed mechanisms of resistance to 5-fluorouracil include defective drug uptake, altered anabolic, and/or catabolic enzyme activity, and multiple mechanisms revolving around thymidylate synthase in respect to its substrate binding, gene amplification, and mutations [100,109,110,111,112,113,114,115,116]. Additional studies are needed to dissect the precise mechanism by which mGluR4 contributes to the development, progression, and resistance of colorectal cancer to 5-fluorouracil.

### 4.3. Glioma

The role of glutamatergic signaling is well established in glioma development and progression. Glioma arises from glial cells, a broad class of non-neuronal cells that regulate the functions and metabolic activities of neuronal cells [117,118]. Glial cells comprise ependymal cells, oligodendrocytes, and astrocytes [118]. Several studies have shown that mGluR1, mGluR2, or mGluR3 signaling are linked to the increased tumorigenicity and metastatic potential of gliomas through activation of the MAPK and PI3K/AKT pathways (Table 1) [88,91,93,119]. Gliomas can be fast-growing or slow-growing, and this classification correlates with response to treatments [120]. In glioblastoma, there are high levels of circulating glutamate that stimulate mGluRs to support tumor progression [91]. Furthermore, riluzole treatment reduces the aggressiveness of glioma cells [119,121]. One of the known functions of riluzole is the inhibition of glutamate release into the extracellular environment. However, the upregulation of EAAT2, a reuptake glutamate transporter, may be another target of riluzole in glioma cells [122]. This phenomenon parallels that in ALS, where riluzole treatment upregulates EAAT2 expression in glial cells to clear the synaptic glutamate [122,123]. The inhibition of mGluR1 with the silencing of RNA was shown to reduce the viability, invasiveness, and migratory activities of human glioma cells (U87) in vitro, and reduced U87 tumor progression in vivo [88]. Supporting evidence has shown that members of the mGluR family are essential players in glioma pathogenesis, as shown by the targeting of group II mGluRs (mGluRs 2/3) with LY341495, a group II antagonist, or riluzole, which reduced the aggressiveness of gliomas [91,93]. In addition, Khan and colleagues have demonstrated that treating mGluR3-expressing U87 glioma cells with riluzole increased DNA damage and the cytotoxicity of glioma cells both in vitro and in vivo; furthermore, treatment with riluzole sensitizes U87 cells to γ-radiation [119].

### 4.4. Kidney Cancer

The introduction of *Grm1* cDNA into immortalized primary baby mouse kidney (iBMK) cells led to cell transformation in vitro and tumorigenesis in vivo [89]. Similar to melanoma and breast epithelial cells, the sustained functional expression of mGluR1 is required for iBMK cells to maintain their transformed and tumorigenic characteristics in vitro and in vivo [89]. mGluR1 expression was detected in renal cell carcinoma (RCC) cell lines and biopsies [89]. Elevated glutamate levels were detected in conditioned cultured media [89]. The inclusion of riluzole led to a reduction in cell growth in vitro and tumorigenesis in vivo, suggesting that mGluR1 signaling participates in RCC development and/or progression (Table 1) [89]. Genetic variations of *GRM3* and *GRM4* in RCC are correlated with worse survival, while *GRM5* is a risk factor for developing RCC [124].

### 4.5. Melanoma

Prickett and colleagues performed a GPCR-targeted mutational analysis on melanoma samples and identified 755 potential somatic mutations in 734 GPCR genes [125]. They found that *GRM3*, the gene encoding mGluR3, was one of the most frequently mutated genes [125]. The identification and characterization of the mutational hotspot p.Glu870Lys in *GRM3* suggests its functional importance in melanoma tumorigenesis [125]. Three other somatic mutations were identified and characterized based on their positions within functional domains of *GRM3* [125]. Genetic manipulations in several melanoma cell lines by either the knockdown of endogenously mutated *GRM3* or the introduction of lentiviruses with identified mutations of *GRM3* into melanoma cells with wild type endogenous *GRM3* led to decreased or increased cell proliferation and migration respectively, suggesting the mutated *GRM3* to be a driver in melanoma pathogenesis [125]. Ceol and co-workers demonstrated that the expression of a mutated *GRM3* in zebrafish resulted in melanosome aggregation in the cell body and deregulated the cAMP signaling that mediates melanosome trafficking [126]. These results suggest that enhancing cAMP signaling may potentially be a therapeutic approach to treating mGluR3-expressing melanomas [126,127].

The expression of mGluR5 has been detected in melanocytes and melanoma cells [54,96,128]. Choi and colleagues demonstrated that the overexpression of mGluR5 could transform melanocytes into malignant melanomas with 100% penetrance and hyperactivation of the MAPK pathway [128]. Our group has shown earlier that the complete knockout of *GRM5* did not alter melanoma pathogenesis driven by the ectopic expression of mGluR1-mediated melanomagenesis [54].

Taken together, these data show that mGluR-mediated melanomagenesis could be derived from overexpression, aberrant expression, or mutations within *GRMs*, and direct them to function as oncogenes in the context of melanomas (Table 1).

### 4.6. Oral Squamous Cell Carcinoma

Oral cancer is categorized based on the region of the mouth in which the neoplasm arises. Oral Squamous Cell Carcinoma (OSCC) arises from the oral cavity, pharyngeal regions, and salivary glands, and accounts for 90% of all oral neoplasms [129]. It was shown that mGluR5, but not mGluR1, can serve as a prognostic marker for OSCC [98]. mGluR5 expression increases during OSCC progression, which corresponds with a lower 5-year survival rate compared to OSCC patients without mGluR5 expression [98]. Using an mGluR5 agonist, DHPG, and an mGluR5 antagonist, MPEP, Choi and colleagues showed that mGluR5 signaling enhances metastasis, invasion, and adhesion in vitro, supporting its role as an oncogene (Table 1) [98]. The epigenetic regulation of *GRM5* may also be involved in OSCC [130].

### 4.7. Osteosarcoma

Osteosarcoma is a malignant bone tumor that typically affects children and teenagers [131]. In osteosarcoma, the overexpression of mGluR4 led to decreased cellular proliferation, migration, and invasion (Table 1) [131]. Liao and colleagues showed the establishment of autocrine loops in mGluR5-expressing osteosarcoma cells, SaOS-LM7, where glutamate is released into the tumor microenvironment, which subsequently activates the mGluR5 receptor on these cells to support tumor growth (Table 1) [132]. The blockade of mGluR5 signaling through pharmacological inhibitors (riluzole or Fenobam, a negative allosteric modulator of mGluR5) or by genetic means resulted in a decrease in osteosarcoma cell growth, reduced tumor motility, and the upregulation of apoptosis [132].

### 4.8. Pancreatic Cancer

mGluR1 expression was detected in pancreatic cancer and shown to be regulated by the long non-coding RNA (lncRNA) HOXA distal transcript antisense RNA (HOTTIP) [133,134,135]. lncRNAs are byproducts of RNA polymerase II transcription, which are now understood to be important for transcriptional regulation, and this has been shown to be associated with the development of cancer [133,134,135]. In pancreatic cancer, the connection between mGluR1 and HOTTIP was suggested to be a potential prognostic marker [133,136]. Altered levels of HOTTIP are correlated with changes in mGluR1 expression, and these changes were linked to tumor cell viability, survival, migration, invasion, and apoptosis [133,136]. Reduced HOTTIP expression in pancreatic cells was shown to induce apoptosis. It will be interesting to explore whether HOTTIP is also involved in the regulation of mGluR1 in other cancers.

### 4.9. Prostate Cancer

mGluR1 was shown to be a relevant prognostic biomarker for prostate cancer and has been shown to be involved in the growth, migration, and invasiveness of prostate cancer cells [97,99,137]. Normal prostate cells express little to no mGluR1, while primary and metastatic prostate cancer cells overexpress mGluR1 [137]. In late-stage prostate cancer patients, higher serum glutamate levels have been detected [97,137]. These results suggest that blocking glutamatergic signaling in prostate cancer could be a viable therapeutic approach. This notion was examined by Vessella and colleagues, who treated mGluR1-expressing prostate cancer cells with riluzole, which led to the induction of apoptosis and the reduced metastatic capabilities of these cells, as well as inhibition of tumor growth (Table 1) [97]. Ali and colleagues described single nucleotide polymorphisms of *GRM1* that may affect splicing, ligand binding, and downstream signaling in prostate cancer cells [138,139]. Additional investigations are needed to assess the consequences of these single nucleotide polymorphisms in *GRM1* in prostate cancer development and progression.

### 4.10. mGluRs in T-Cell Biology and T-Cell Cancers

The influence of glutamate on T-cells depends on the specific mGluR being expressed on the T cell subtypes, the resting or active state of T-cells, and the presence or absence of other concurrent stimuli [140]. In general, glutamate was shown to be involved in numerous functions of T-cells, including T-cell activation, survival, adhesion, migration, proliferation, and the inhibition of antigen-induced apoptosis [140]. It has been proposed that tumor immune evasion in T-cell cancers is mediated through the elevated glutamate levels in mGluR-expressing cells, which promote tumor progression via the hyperactivation of mGluRs on tumor cells, together with mGluR activation on normal T-cells that reduces the expansion of cytotoxic T-cells within the tumor microenvironment [140]. Group I (mGluR1/5), group II (mGluR2/3), and group III (mGluR4/6/7/8) mGluRs are expressed in human T-cell leukemia (Jurkat, FRO, and SUP-T1) and T-cell lymphoma (HUT-78, and H9) cell lines [86,140,141].

**Table 1 cells-11-02857-t001:** *GRMs* function as an oncogene or tumor suppressor in various cancers.

Cancers	*GRMs*	Oncogene/Tumor Suppressor	Reference
Breast Cancer	*GRM1*, *GRM8*	Oncogene	[105,106,107]
	*GRM4*	Tumor Suppressor	[108]
Colorectal Carcinoma	*GRM4*	Oncogene	[100]
Glioma	*GRM1*, *GRM2*, *GRM3*	Oncogene	[88,91,93,119]
Kidney Cancer	*GRM1*	Oncogene	[89,138]
	*GRM3*, *GRM4*, *GRM5*	Function Unknown	[124]
Melanoma	*GRM1*, *GRM3*, *GRM5*	Oncogene	[38,125,126,128]
Oral Squamous Cell Carcinoma	*GRM5*	Oncogene	[98]
Osteosarcoma	*GRM4* *GRM5*	Tumor Suppressor Oncogene	[131,132]
Pancreatic Cancer	*GRM1*	Function Unknown	[133,136]
Prostate Cancer	*GRM1*	Oncogene	[97]
T-Cell Cancers	*GRM1-GRM8*	Function Unknown	[86,140,141]

The specific oncogenic activities of these mGluRs in T-cell leukemia and lymphoma are largely unknown, and needs further investigation; however, the involvement of ionotropic Glutamate Receptors (iGluRs) has been documented well in T-cell cancers. For reviews, please see these citations [140,141]. Taken together, the results from these studies support the notion that different mGluRs have different inter- and intra-cancer functions.

## 5. Altered Glutamate Metabolism in Cancers

Altered metabolism in cancer cells is a direct result of modifications in intracellular signaling pathways that stem from mutations in genes. These modulated pathways enable cancer cells to escape normal cellular regulations while maintaining the high demand for energy and perpetual tumor proliferation. Metabolic activities in untransformed normal cells, under aerobic conditions, rely primarily on the cell’s ability to undergo glycolysis via mitochondrial oxidative phosphorylation (OXPHOS) to generate energy in the form of adenosine triphosphate (ATP) [142,143]. Hypoxia, a common characteristic of solid tumors, occurs due to the rapid growth of tumor cells and their inability to sustain oxygenation, as a result of this, a shift from OXPHOS to aerobic glycolysis occurs. Aerobic glycolysis, also known as the “Warburg effect”, results in a large production of lactate. This phenomenon was first observed by the Nobel laureate Otto Warburg in 1924 [142,144,145].

The metabolic rewiring of many tumor cells to glutamine metabolism over other non-essential amino acids dominates many cancer cells. All cells require glutamine for growth and survival, but cancer cells are much more sensitive to glutamine deprivation, a phenomenon known as “glutamine addiction” [146,147]. Experimental evidence suggests that glutamine is the major respiratory fuel for energy production in tumor cells [148]. In addition to being a nitrogen donor for protein and nucleotide synthesis, glutamine supports the replenishment of the mitochondrial carbon pool through a process known as anaplerosis [147]. It has been demonstrated that tumor cells can route glutamine through the tricarboxylic acid (TCA) cycle in reverse via a pivotal mitochondrial glutaminolytic amidohydrolase enzyme known as glutaminase (GLS) [149].

Glutaminase converts glutamine to glutamate via the hydrolytic cleavage of glutamine in the first step of reductive metabolism. Conversely, the reverse conversion of glutamate back into glutamine is catalyzed by another metabolic enzyme, glutamine synthetase (GS), via the ligation of glutamate and ammonia. GS has been implicated in cancers, such as primary liver cancer and hepatocellular carcinoma [147,150,151]. One of the possible ways for cancer cells to obtain enough glutamate is by elevating the influx of glutamine into cells followed by enhanced GLS activity. GS, on the other hand, may contribute to catabolism, fueling the tricarboxylic acid (TCA) cycle through coupling with GLS [152,153,154]. Numerous studies imply that the upregulation of GLS plays a critical role in tumor proliferation in various types of cancers, including glioma, lymphoma, non-small cell lung cancer, prostate cancer, and TNBC [155,156,157,158]. Furthermore, the downregulation or inhibition of GLS has been shown to reduce the proliferation of these tumors [158,159]. GLS inhibition has also been shown to enhance the effectiveness of chemotherapy, reduce metastatic progression, and improve the efficacy of other targeted therapies, suggesting the critical role of targeting GLS in improving overall patient response [160,161,162,163,164]. The overexpression of GLS has been shown to play a vital role in upregulating glutamine catabolism by supporting the production of molecules essential for tumor growth and proliferation This fundamental insight that basic research has provided into the understanding of the glutaminolysis pathway has allowed for the development of various GLS inhibitors.

The regulation of GLS in cancer remains to be fully elucidated. Several studies have proposed different mechanisms by which GLS is regulated in cancer cells (Figure 4). It was previously demonstrated that mTOR complex 1 (mTORC1) is a positive regulator of GLS expression through the c-Myc axis [165,166]. Importantly, mTORC1 has been shown to act as a critical molecular link between growth signals and the processes underlying anabolic cell growth and differentiation [167]. Recent findings in human ductal pancreatic adenocarcinoma (PDAC) specimens reveal the critical regulation of GLS by succinate-CoA ligase ADP-forming subunit beta (SUCLA2)-coupled GLS succinylation during oxidative stress (Figure 4) [168]. In breast cancer cells, sirtuin-5 (SIRT5), a mitochondrial NAD^+^-dependent lysine deacylase, was shown to stabilize GLS by desuccinylating GLS and preventing it from ubiquitin-mediated degradation [169]. In addition, increased SIRT5 expression in human breast tumors appears to associate with poor patient prognosis [169]. These results put forth a notion of how SIRT5 rewires metabolisms in breast cancer. GLS has two isoforms: GLS1 is expressed in many types of cancer, while GLS2 is primarily expressed in the liver [170]. The specific function of the general control of amino acid synthesis 5 like 1 (GCN5L1) is speculated to regulate mitochondrial protein acetylation, which includes both forms of GLS and has been shown to modulate glutaminolysis to favor the development of hepatocarcinoma [171,172]. Other studies have shown a direct relationship between c-Myc and GLS expression—Gao and colleagues reported that c-Myc transcriptionally represses miR-23a and miR-23b, leading to the higher expression of GLS, which upregulates the glutamine catabolism in human P-493B lymphoma cells and PC3 prostate cancer cells [173]. A recent study implicates the role of a novel glutaminase antisense lncRNA (GLS-AS)-mediated Myc/GLS pathway in a pancreatic cancer model (Figure 4) [174]. Preliminary studies conducted in melanoma cells by our group have shown that the knockdown of c-Myc via shRNA did not lead to a parallel downregulation of GLS expression, suggesting that a concomitant decrease in mTORC may be essential. Others have postulated the notion that a simultaneous suppression of mTORC and c-Myc is necessary to perceive a subsequent decrease in GLS expression [165,166,175]. Our group has established that GLS overexpression, in mGluR1-expressing melanoma cell lines transpires at least in part through the mTORC/c-Myc axis as seen through the steady knockdown of c-Myc with reduced mTOR phosphorylation and the subsequent downregulation of GLS (unpublished results) [68]. Both rapamycin and everolimus (mTORC1 inhibitors) have displayed inhibitory effects on the growth, proliferation, and survival of tumors, including melanoma, with minimal toxicity [176]. These results are crucial to understanding the underlying molecular mechanisms of oncogenic alterations in the glutaminolytic activity of cancer cells. These novel findings, combined with our data on melanoma, may aid in our understanding of these regulatory mechanisms and could possibly help expand opportunities for novel drug therapies, as well as optimizing current treatments. Please note that Figure 4 summarizes some of the recently published reports on the proposed mechanisms by which GLS maybe regulated in some cancers. It is possible that one or more of these mechanisms may also apply in mGluR1-expressing melanomas.

## 6. Discussion and Future Directions

mGluRs have been extensively studied in the CNS, with vast numbers of agonists and antagonists having been developed to modulate mGluR function/activity. These studies have provided a strong foundation for our understanding of mGluR functions and created numerous potential candidates for targeting mGluR signaling in cancer. A systematic high-throughput screen of these agonists/antagonists may uncover potential novel therapeutic approaches to cancer. Our findings on elevated extracellular glutamate in conditioned media of mGluR1-expressing tumor cells led us to examine known compounds that modulate the export of glutamate and functionally regulate the formation of glutamate from glutamine. We showed that the inclusion of riluzole, an inhibitor of glutamate export, and/or CB-839, an inhibitor of glutaminase, in the growth media led to a reduction in cell growth and subsequent tumor progression in vitro and in vivo [39,68]. The application of riluzole has shown promise in melanoma, breast cancer, glioma, prostate cancer, and kidney cancer [39,89,97,101,119,121,177,178,179]. Our lab and others have provided pre-clinical data that support the combination of riluzole/troriluzole with different therapeutic approaches that target different oncogenic pathways: oxidative stress/genomic instability (γ-radiation therapy or PARP inhibitors), cell metabolism (CB-839), and angiogenesis/metastasis (anti-VEGF treatments or inhibitors for exosome biogenesis and secretion) [28,57,68,69,119,178,179,180,181,182,183].

To further expand our knowledge of the possible roles of various classes of mGluRs in cancer development and progression, one could first use the available public database to identify the possible involvement of mGluRs in various cancers and then use cultured cell lines to perform the initial assessments using pharmacological agonists/antagonists, followed by genetically manipulating the expression of mGluRs and analyzing the consequences via such strategies. Using the available human tissue specimens and histological slides, one could further evaluate the human relevance of such examinations. Using human 3D organoid cultures could be a relatively feasible approach for this investigation [184]. Finally, in vivo approaches, including the establishment of PDXs from human tissues and genetically engineered experimental mouse model systems, can be used to validate in vitro findings. Our lab and others have conducted similar types of studies, which have helped improve our understanding of the modes of action and functions of mGluR1 in melanoma development and progression [38,51,54,56].

## Figures and Tables

**Figure 1 cells-11-02857-f001:**
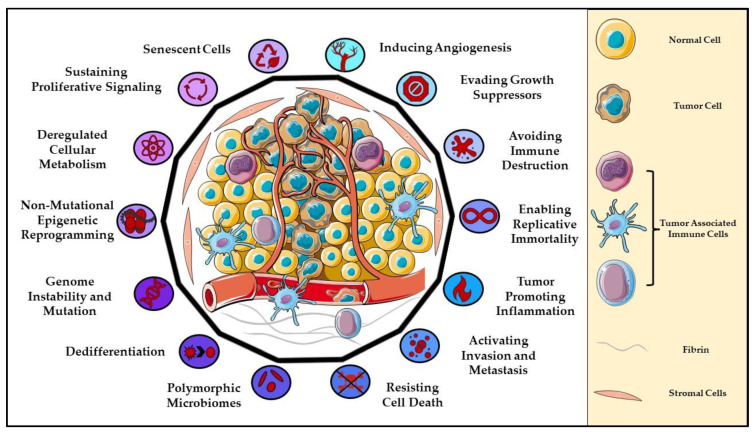
Hallmarks of Cancer. This adapted illustration displays the fourteen Hallmarks of Cancer put forth by Hanahan and Weinberg [9,10,11]. The inner polygon shows the tumor microenvironment, and the outer circles highlight each cancer hallmark with a unique symbol. The color gradients used for the outer circles are to show difference between each hallmark. The symbols are generic and have been selected based on the scientific terminology used for each hallmark; however, they are not used globally to represent these hallmarks. Parts of the figure were created by using pictures from Servier Medical Art. Servier Medical Art by Servier is licensed under a Creative Commons Attribution 3.0 Unported License (https://creativecommons.org/licenses/by/3.0/, accessed on 27 May 2022).

**Figure 2 cells-11-02857-f002:**
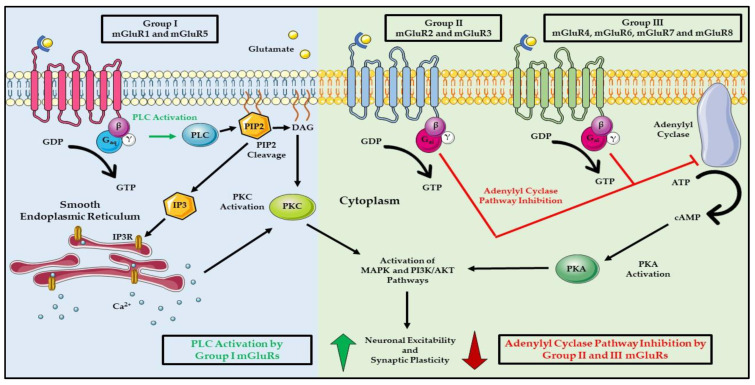
Modes of action of mGluRs. The known functions of Group I, II, and III mGluRs upon activation by the natural ligand, glutamate, with the downstream PLC and adenylyl cyclase signaling pathways. The straight black arrows indicate the cellular signaling pathway. The green arrows indicate activation process, and the red arrows indicate inhibition process. The curved black arrows indicate conversion of one molecule to another. Abbreviations: metabotropic glutamate receptors (mGluR: protein; *Grm*: mouse gene; *GRM*: human gene), guanine diphosphate (GDP), guanine triphosphate (GTP), phospholipase C (PLC), phosphatidylinositol-4,5-diphosphate (PIP_2_), diacyl-glycerol (DAG), inositol 1,4,5-triphosphate (IP_3_), inositol 1,4,5-triphosphate receptor (IP3R), protein kinase C (PKC), adenosine triphosphate (ATP), cyclic adenosine monophosphate (cAMP), and protein kinase A (PKA). Parts of the figure were created using pictures from Servier Medical Art. Servier Medical Art by Servier is licensed under a Creative Commons Attribution 3.0 Unported License (https://creativecommons.org/licenses/by/3.0/, accessed on 29 May 2022).

**Figure 3 cells-11-02857-f003:**
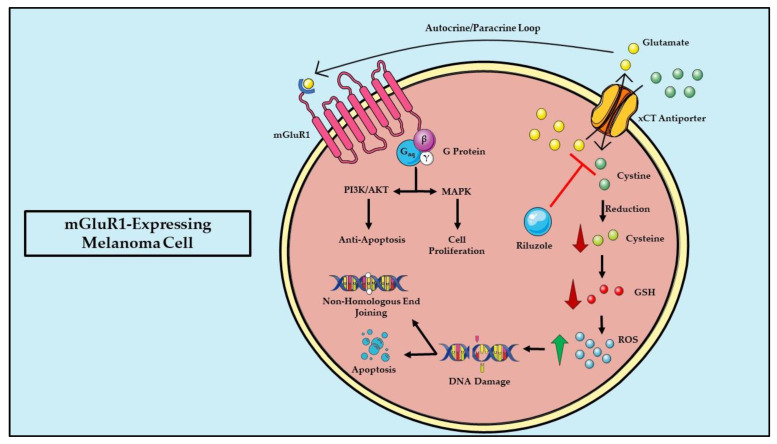
The proposed mechanism of action of riluzole in mGluR1-expressing melanoma cells. This illustration highlights the mechanism of cystine import, and glutamate export into cells via the xCT antiporter, as well as the proposed mechanism of action of riluzole in mGluR1-expressing melanoma cells via blockade of xCT activity. The straight black arrows indicate the cellular signaling pathways. The red arrow indicates the inhibitory activity of riluzole on the xCT antiporter. The curved black arrows indicate transport of molecules inside and outside of the cell. Abbreviations: glutathione (GSH) and reactive oxygen species (ROS). Parts of the figure were created using pictures from Servier Medical Art. Servier Medical Art by Servier is licensed under a Creative Commons Attribution 3.0 Unported License (https://creativecommons.org/licenses/by/3.0/, accessed on 15 June 2022).

**Figure 4 cells-11-02857-f004:**
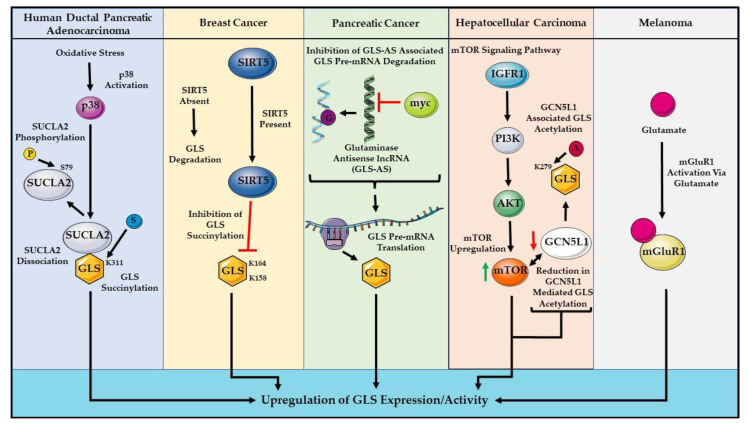
Known regulators of GLS expression/activity in different cancers. Symbols P, S, A, and G represent phosphate, succinyl, acetyl group, and GLS-AS protein, respectively. The black arrows indicate the series of events occurring in the cellular signaling pathways in various cancer types leading to the upregulation of GLS expression/activity. The red arrow next to GCN5L1 indicates a decrease in activity. The green arrow next to mTOR indicates upregulation. Brackets have been used to connect two different interdependent pathways. Abbreviations: succinate-CoA ligase ADP-forming subunit beta (SUCLA2), glutaminase (GLS), sirtuin-5 (SIRT5), glutaminase antisense lncRNA (GLS-AS), insulin-like growth factor receptor 1 (IGF-R1), and general control of amino acid synthesis 5 like protein 1 (GCN5L1). Parts of the figure were created using pictures from Servier Medical Art. Servier Medical Art by Servier is licensed under a Creative Commons Attribution 3.0 Unported License (https://creativecommons.org/licenses/by/3.0/, accessed on 9 June 2022).

## Data Availability

Not applicable.

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
