# Peer review of "Implications of a Neuronal Receptor Family, Metabotropic Glutamate Receptors, in Cancer Development and Progression"

_cells, 2022, doi:10.3390/cells11182857_

Round 1

Reviewer 1 Report

This is an excellent review on   Metabotropic Glutamate Receptors and cancer. The review is written by a group of scientists who originated that line of research, which also led to clinical trials (riluzole).

 A list of abbreviations would be helpful to understand the text. Abbreviations  could be also explained  in captions under the drawings: e.g. Fig . 3  GSH (glutathione) is not explained

Are there any data on GS (glutamine synthetase) inhibitors and cancers?

Reviewer 2 Report

The authors provide a detailed review on the current knowledge of metabotropic glutamate receptors (mGluRs) in cancer. Although there are quite some description about the work by the authors' lab. The overall review is balanced with sufficient citations of work by others. The main issue is the English language, especially the misuse of punctuation that makes the ms difficult to read. It is hard to list all the problem here, so I decided to mark all areas in the attached PDF file. Hopefully, these will be helpful. I do want to emphasize that not every "and" needs to be preceded with a comma. This has been a problem throughout the entire ms with few exceptions.

Another problem is section 5 on Metabotropic Glutamate Receptor and Altered Glutamate Metabolism. This section talks about GLS and glutamine, but not mGluRs, except for one mention of the mGluR1 expressing melanoma cell line. It is not clear why this section is included in this ms. If the purpose is to provide one potential mechanism of glutamate production and release in tumors, it needs to be stated clearly. However, is this the actual and main reason for glutamate production in cancer cells?

The authors seem to have no appreciation of Gβγ as transducers of heterotrimeric G protein signaling. The effects of Gβγ should be considered especially when discussing Gi/o-coupled receptors.  For this reason, it is also not a good idea to refer the G proteins activated by mGluRs as just Gα.

Reviewer 3 Report

Thank you for drafting this nice review. I had a number of questions and comments upon reading the manuscript.

Line 155-.. It is unclear from the text what exactly was taken from adipose tissue to the fibroblasts, and what de DNA clones relate to. Maybe it can be rephrased so the reader does not need to refer to the citation.

Line ~270. The text describes oxidative damage due to Riluzole blocking the xCT port. Is anything known about such effects of Riluzole when used in other conditions (e.g. ALS) ? What is different in the context of Melanoma?

Line 288 Not a fan of "to name a few" three times in the text.

Line 299 Please review the citations, the sentence mentions Speyer but discusses and cites Bastiaansen. In the next line "they" does relate to Speyer it seems. 

Line 332 resistancce

Line 358 evidence has

Section 5, Figure 4. This section discusses many differences in the way GLS is regulated. It would be interesting if the authors could achieve a bit more of an integrated analysis. Fig 4 mentions SIRT5 pathway in breast cancer. Breast cancers also often express mGluR1, could the pathway discussed for that also be important there? The text discusses mTORC1 mediation in mGluR1 expressing melanoma, would that also apply in Breast cancer, and should the Melanoma column of Figure 4 be supplemented with mTORC information? The pathway for Melanoma is quite unpopulated in the figure. The section title is "Metabotropic Glutamate receptor and Altered Metabolism" but mentions only once, briefly,  a mGluR (1). 

Round 2

Reviewer 2 Report

The authors made extensive effort to revise and improve the ms. Most of the revisions are reasonable, but I have trouble understanding why in quite a few cases, they changed "is" to "are" even through it is obvious that the subject noun is in the singular form, e.g., lines 79, 148, 150, 157, 513, etc. In some of these cases, the subject noun could be changed to plural. In some other cases, it is necessary to correctly identify the subject noun, which is not always the one immediately preceding the verb.

Although I still do think that the authors provide a convincing argument as to why GLS is discussed in this review article, I have my own reasons to believe that GLC could be a relevant topic here. Therefore, I recommend accepting the paper in its current form, noting that grammars need to be carefully checked.